# Scale Invariance of Graph Neural Network for Node Classification

## Abstract

We address two fundamental challenges in Graph Neural Networks (GNNs) for node classification: (1) the lack of theoretical support for invariance learning, a critical property in image processing, and (2) the absence of a unified model capable of excelling on both homophilic and heterophilic graph datasets. To tackle these issues, we establish and prove scale invariance in graphs, extending this key property to graph learning, and validate it through experiments on real-world datasets. Leveraging directed multi-scaled graphs and an adaptive self-loop strategy, we propose ScaleNet, a unified network architecture that achieves state-of-the-art performance across four homophilic and two heterophilic benchmark datasets. Furthermore, we show that through graph transformation based on scale invariance, uniform weights can replace computationally expensive edge weights in digraph inception networks while maintaining or improving performance. For another popular GNN approach to digraphs, we demonstrate the equivalence between Hermitian Laplacian methods and GraphSAGE with incidence normalization. In sum, ScaleNet bridges the gap between homophilic and heterophilic graph learning, offering both theoretical insights into scale invariance and practical advancements in unified graph learning. Our implementation is publicly available at https://anonymous.4open.science/r/ScaleNet-B663.

## 1. Introduction

Graph Neural Networks (GNNs) have emerged as powerful tools for learning from graph-structured data, with significant applications in node classification tasks such as protein function prediction (Gligorijević et al., 2021), user catego-

rization in social networks (Hamilton et al., 2017), Internet content recommendation (Ying et al., 2018), and document classification in citation networks (Kipf & Welling, 2016). Despite their proven effectiveness in node classification, GNNs face two major limitations that hinder their theoretical understanding and practical deployment.

First, from a theoretical perspective, GNNs lack robust theoretical foundations for invariant learning—a fundamental concept well-established in image classification tasks. While Convolutional Neural Networks (CNNs) leverage invariance properties to enable effective data augmentation through image transformations, GNNs lack analogous theoretical guarantees. This limitation is critical for node classification, where predictions should remain invariant across different neighborhood scales—from immediate neighbors to nodes multiple hops away. The absence of a rigorous framework for graph invariance not only limits our theoretical understanding but also impedes the development of robust GNN architectures.

Second, from an empirical standpoint, existing GNN architectures demonstrate a notable dichotomy in their node classification performance: they typically excel either on homophilic graphs (Tong et al., 2020a) (where connected nodes share similar labels) or heterophilic graphs (Rossi et al., 2024) (where connected nodes have different labels). This dichotomy raises important questions about the underlying mechanisms that determine model effectiveness across different graph types.

To address these limitations, we make three key contributions, all of which are **constructive**:

1. We establish and prove scale invariance in graphs, extending this fundamental concept from image processing to graph learning.

2. We develop a unified network architecture that translates this theoretical insight into practice.

3. We introduce an adaptive self-loop strategy that dynamically adjusts to graph homophily characteristics.

In addition, our technical analysis reveals two **destructive insights** that simplify existing approaches without compromising performance:

[1]Anonymous Institution, Anonymous City, Anonymous Region, Anonymous Country. Correspondence to: Anonymous Author <anon.email@domain.com>.

Preliminary work. Under review by the International Conference on Machine Learning (ICML). Do not distribute.

- By applying graph transformation based on scale invariance, uniform weights can replace the computationally expensive edge weights in digraph inception networks (Tong et al., 2020a;b), maintaining or even improving performance while reducing complexity.

- There is an equivalence between Hermitian Laplacian methods (e.g., MagNet (Zhang et al., 2021)) and Graph-SAGE (Hamilton et al., 2017) when incidence normalization is applied. This is proved in Appendix C.2.

Our evaluation shows that the proposed method achieves state-of-the-art results on four homophilic and two heterophilic graphs. Compared to existing approaches, our method offers superior performance on homophilic datasets compared to Dir-GNN (Rossi et al., 2024), better handling of heterophilic data than MagNet (Zhang et al., 2021), and improved efficiency over real symmetric Laplacian methods. Furthermore, our multi-scale graph approach provides notable advantages for highly imbalanced datasets through implicit data augmentation.

## 2. Unnecessary Complexity: A Case for Simplification

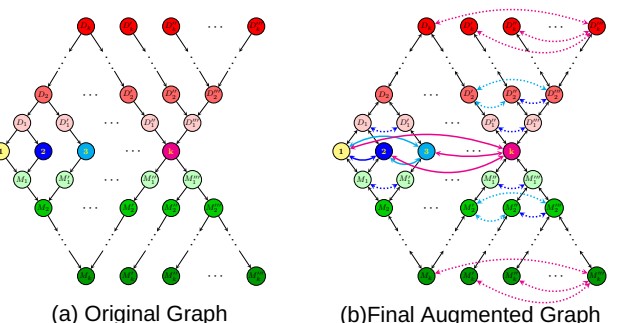

(a) Original Graph      (b)Final Augmented Graph

Figure 1: Edge augmentation by stacking multi-scale graphs in Digraph Inception Model.

State-of-the-art GNNs for homophilic graphs include Digraph Inception Networks such as DiGCN(ib)[1] (Tong et al., 2020a) and SymDiGCN (Tong et al., 2020b), which use higher-order proximity for multi-scale features. However, their reliance on random walks makes edge weights across scales crucial. DiGCN(ib) requires computationally expensive eigenvalue decomposition to determine these weights, whereas SymDiGCN relies on costly node-wise outer product computations. These computational requirements pose significant scalability challenges, particularly for large-scale graph applications.

---

[1]In this paper, DiGCN interchangably with DiG, DiGCNib interchangably with DiGib, DiGi2

Their success stems from edge augmentation through various proximities, as shown in Figure 1.

Table 1: Performance of Inception models on the Telegram dataset. "BN" indicates the addition of batch normalization to the original model. The **R**iG(ib) model assigns random weights in uniform distribution to edges within the range [0.0001, 10000], and The **1**iG(ib) model assigns weight 1 to all scaled edges.

| Model | No BN | BN | Model | No BN | BN |
|---|---|---|---|---|---|
| DiG | 67.4±8.1 | 63.0±7.6 | DiGib | 68.4±6.2 | 77.4±5.1 |
| **1**iG | 86.0±3.4 | 95.8±3.5 | **1**iGib | 86.2±3.2 | 94.2±2.7 |
| **R**iG | 85.2±2.5 | 91.0±6.3 | **R**iGib | 86.4±6.2 | 86.4±6.6 |

Instead of computing edge weights for higher-scaled edges, we replace the edge weights in DiGCN(ib) with uniform weights of **1**, resulting in our simplified models (**1**iG, **1**iGi2).

We show that the computational cost associated with DiGCN(ib)'s edge weights is unnecessary, as replacing them with uniform weights of **1** still yields competitive results. Further experiments with random weights (**R**iG(i2)) in range [0.0001, 10000] show even random weighting outperforms DiGCN(ib) (Table 1), both with and without batch normalization. In Appendix C.1, we provide an explanation for why the edge weights generated by DiGCN(ib) perform worse than random weights.

Notably, that **1**iGib chieves comparable performance to DiG(ib) holds consistently across all 15 datasets we tested (Table 2). More details about the datsets are shown in Appendix E.3. Additionally, we replaced the edge weights in SymDiGCN (Tong et al., 2020b) with uniform weights of **1**, resulting in our simplified models **1**ym, whose performance is comparable to SymDiGCN across all datasets( Table 4).

This intriguing discovery led us to hypothesize the existence of scale invariance in graphs, where neighborhoods at different hop distances can still be effectively utilized for node classification tasks.

## 3. Scaled Graph

### 3.1. Scaled Ego-Graphs

Let $G = (V, E)$ be a directed graph with $n$ nodes and $m$ edges, represented by an adjacency matrix $A \in \{0, 1\}^{n \times n}$, where $A_{ij} = 1$ indicates the presence of a directed edge from node $i$ to node $j$, and $A_{ij} = 0$ indicates the absence of such an edge. We focus on node classification where node features are organized in an $n \times d$ matrix $X$, where $d$ is the dimension of features and the node labels are $y_i \in \{1, \ldots, C\}$.

**Definition 3.1** (In-Neighbour). An *in-neighbour* of a node $v \in V$ is a node $u \in V$ such that there is a directed edge from $u$ to $v$, i.e., $(u, v) \in E$.

**Definition 3.2** (Out-Neighbour). An *out-neighbour* of a node $v \in V$ is a node $u \in V$ such that there is a directed

Table 2: Ablation study comparing DiG(ib) with two variants: **1**iG(ib) where edge weights are set to 1, and **R**iG(ib) where edge weights are randomly sampled from [0.0001, 10000]. Results on 15 graphs (8 directed, 7 undirected) show that **1**iG(ib) achieves comparable performance to DiG(ib), while **R**iG(ib) occasionally outperforms it (e.g., on Telegram), suggesting that the cost of edge weight computation in DiG(ib) may be unnecessary. Each cell shows accuracy (top) and Macro F1-score (bottom). Entries marked OOM indicate Out of Memory on NVIDIA A40 GPUs with 48GB VRAM. The last two columns provide dataset statistics, with the 'Node' cell showing total nodes (top) and training nodes (bottom).

| Type | Datasets | DiG | DiGib | 1iG | 1iGib | RiG | RiGib | Node | Edge |
|---|---|---|---|---|---|---|---|---|---|
| | **Cornell** | 55.4±7.3 | 69.2±5.4 | 57.0±6.7 | 66.5±7.1 | 44.6±6.9 | 67.8±4.7 | 183 | 298 |
| | F1 | 38.3±7.1 | 49.2±7.1 | 39.4±7.1 | 52.6±11.0 | 25.5±9.6 | 52.4±5.5 | 87 | |
| | **Wisconsin** | 64.7±6.8 | 78.0±6.1 | 64.5±5.3 | 74.7±6.6 | 47.3±6.6 | 72.4±4.8 | 251 | 515 |
| | F1 | 47.0±9.5 | 53.6±8.2 | 42.0±7.7 | 57.6±5.2 | 26.0±6.4 | 61.2±6.6 | 120 | |
| | **Texas** | 62.2±5.1 | 73.0±8.6 | 67.8±5.8 | 70.5±6.2 | 58.6±6.1 | 67.8±9.2 | 183 | 325 |
| | F1 | 38.4±6.4 | 54.4±10.8 | 46.0±7.9 | 55.3±11.3 | 28.7±6.7 | 52.1±12.3 | 87 | |
| | **CiteSeer** | 60.4±2.0 | 66.6±1.5 | 66.6±2.2 | 62.8±2.0 | 52.7±2.4 | 65.5±2.0 | 3,312 | 4,715 |
| **Directed** | F1 | 57.1±1.3 | 62.8±1.8 | 62.6±1.6 | 59.9±1.5 | 49.6±1.7 | 61.1±1.9 | 120 | |
| **Graphs** | **CoraML** | 77.0±1.9 | 76.6±2.1 | 81.0±1.8 | 81.7±1.3 | 79.7±2.5 | 79.5±2.6 | 2,995 | 8,416 |
| | F1 | 76.0±2.0 | 76.2±1.7 | 80.1±1.9 | 80.8±1.3 | 79.0±2.2 | 78.8±2.4 | 140 | |
| | **PubMed** | 74.3±0.6 | 76.9±0.6 | 76.3±0.9 | 76.7±0.2 | 59.0±1.5 | 59.1±1.4 | 19,717 | 44,327 |
| | F1 | 74.0±0.6 | 76.9±0.5 | 76.1±0.8 | 77.0±0.2 | 57.9±1.2 | 58.2±1.4 | 60 | |
| | **WikiCS** | 77.1±1.0 | 78.4±0.6 | 79.1±1.0 | 78.9±0.6 | 73.0±0.5 | 78.6±0.5 | 11,701 | 297,110 |
| | F1 | 74.1±1.0 | 75.8±0.9 | 76.3±0.8 | 76.0±0.9 | 73.5±0.8 | 74.2±0.7 | 580 | |
| | **Telegram** | 76.8±4.5 | 66.0±5.5 | 95.8±3.5 | 93.0±5.1 | 87.2±3.7 | 89.0±4.1 | 245 | 8,912 |
| | F1 | 70.5±6.2 | 64.3±5.5 | 94.7±4.2 | 92.8±4.9 | 85.4±3.9 | 88.6±4.4 | 145 | |
| | **CiteSeer-U** | 69.2±0.6 | 68.9±0.7 | 69.3±0.6 | 68.8±1.0 | 42.0±1.2 | 40.5±5.5 | 3,327 | 4,732 |
| | F1 | 66.1±0.4 | 65.3±0.6 | 66.2±0.5 | 65.5±0.8 | 38.9±1.1 | 38.6±1.9 | 120 | |
| | **Cora** | 79.1±0.7 | 80.8±0.9 | 80.3±1.0 | 80.0±0.7 | 51.5±1.3 | 50.3±2.6 | 2,708 | 5,429 |
| | F1 | 77.3±0.7 | 79.1±0.7 | 78.7±0.8 | 78.4±0.7 | 50.3±1.2 | 51.0±2.1 | 140 | |
| | **PubMed-U** | OOM | OOM | 78.3±0.2 | 77.5±0.4 | 36.8±5.7 | 77.3±0.6 | 19,717 | 108,365 |
| | F1 | OOM | OOM | 78.3±0.1 | 77.4±0.3 | 20.9±3.5 | 77.3±0.6 | 60 | |
| **Undirected** | **CoA-CS** | 91.1±0.4 | 95.1±0.1 | 89.6±0.5 | 95.0±0.1 | 30.4±0.1 | 88.6±0.4 | 18,333 | 163,788 |
| **Graphs** | F1 | 87.7±0.8 | 93.8±0.1 | 84.1±2.5 | 93.7±0.1 | 6.8±0.0 | 86.6±0.7 | 8,793 | |
| | **CoA-Physics** | 95.8±0.2 | 96.8±0.0 | 95.8±0.1 | 96.8±0.0 | 90.9±0.1 | 88.4±0.3 | 34,493 | 495,924 |
| | F1 | 94.4±0.2 | 95.8±0.0 | 94.4±0.1 | 95.7±0.1 | 88.3±0.1 | 84.8±0.6 | 16,555 | |
| | **Photo** | 93.2±0.2 | 91.8±0.3 | 91.8±0.4 | 91.7±0.1 | 28.1±3.3 | 88.4±0.1 | 7,650 | 238,162 |
| | F1 | 91.2±0.3 | 88.6±0.5 | 88.3±1.1 | 88.4±0.2 | 9.2±1.9 | 83.5±0.3 | 3,669 | |
| | **Computers** | 87.5±0.3 | OOM | 89.5±0.3 | OOM | 83.5±0.6 | OOM | 13,752 | 491,722 |
| | F1 | 86.1±0.5 | OOM | 88.7±0.5 | OOM | 82.6±1.2 | OOM | 6,595 | |

edge from $v$ to $u$, i.e., $(v, u) \in E$.

An $\alpha$-depth ego-graph (Alvarez-Gonzalez et al., 2023) includes all nodes within $\alpha$ hops from a central node. We extend this concept to directed graphs and introduce scaled hops, leading to scaled ego-graphs.

**Definition 3.3.** In a directed graph $G = (V, E)$, we define two types of $\alpha$-depth ego-graphs centered at a node $v \in V$.

- **$\alpha$-depth in-edge ego-graph**: $I_\alpha(v) = (V_\leftarrow, E_\leftarrow)$, where $V_\leftarrow$ consists of all nodes that can reach $v$ within $\alpha$ steps, and $E_\leftarrow$ consists of all directed edges between nodes in $V_\leftarrow$ that are within $\alpha$ steps of $v$.

- **$\alpha$-depth out-edge ego-graph**: $O_\alpha(v) = (V_\rightarrow, E_\rightarrow)$, where $V_\rightarrow$ consists of all nodes that can be reached from $v$ within $\alpha$ steps, and $E_\rightarrow$ consists of all directed edges from $v$ to nodes in $V_\rightarrow$ within $\alpha$ steps.

As illustrated in Figure 2, a 1-depth ego-graph for an undirected graph includes nodes labeled I (in-neighbor) and O (out-neighbor). In the case of a directed graph, the 1-depth in-edge ego-graph comprises nodes labeled I along with the center node and all the edges connecting them, whereas the 1-depth out-edge ego-graph comprises nodes labeled O along with the center node and all the edges connecting them.

The 1-hop neighbour with different adjacency matrix is shown in Table 3.

Table 3: 1-hop neighbours for GNN with different adjacency matrices

| Adj. Matrix | $A$ | $A^T$ | $AA$ | $A^T A^T$ | $AA^T$ | $A^T A$ |
|---|---|---|---|---|---|---|
| 1-hop Neighb. | I | O | II | OO | IO | OI |

**Definition 3.4.** A **scaled-edge** is defined as an ordered sequence of multiple directed edges, where the **scale** refers to the number of edges in this sequence. Specifically, a $k^{th}$-**scale edge** is a scaled-edge composed of $k$ directed edges, also referred to as a $k$-**order edge**.

An $\alpha$-**depth scaled ego-graph** includes all nodes that are reachable within $\alpha$ hops of scaled-edge from a given center node.

A $1^{st}$-scale edge, includes in-edge (I) and out-edge (O), connecting to in-neighbor (I) and out-neighbor (O) nodes, respectively, as shown in Figure 2. Considering a $2^{nd}$-scale edge, there are four types: II, IO, OI, and OO, each connecting to nodes labeled in Figure 2 accordingly.

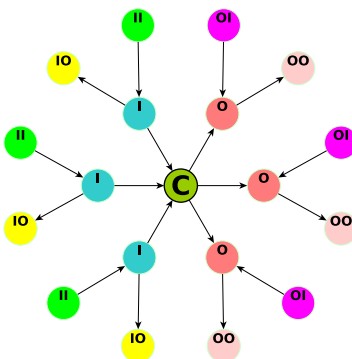

Figure 2: An illustration of scaled ego-graphs. For directed graphs, the 1-depth in-edge ego-graph comprises nodes labeled "I" along with the center node "C" and all in-edges between them, whereas the 1-depth out-edge ego-graph comprises nodes labeled O along with the center node and all out-edges between them. The four types of 1-depth $2^{nd}$-scaled ego-graphs are composed of nodes labeled "IO", "OI", "II", and "OO", with the center node and all corresponding $2^{nd}$-scaled edges between them.

### 3.2. Scale Invariance of Graphs

The concept of scale invariance, well-known in image classification as the ability to recognize objects regardless of their size, can be extended to graphs. In the context of node classification, each node to be classified can be viewed as the center of an ego-graph. Thus, for node-level prediction tasks on graphs, each input instance is essentially an ego-graph $G_v$ centered at node $v$, with a corresponding target label $y_v$. Scale invariance in graphs would imply that the classification of a node remains consistent across different scaled ego-graphs.

**Definition 3.5.** Let $S_k$ denote the set of all $k^{th}$-scale edges and $G_\alpha^k(v)$ denote the set of all $\alpha$-depth $k^{th}$-scale ego-graphs centered at node $v$. Then we have the following equations:

$$S_k = \{e_1 e_2 \dots e_k \mid e_i \in \{\rightarrow, \leftarrow\}, 1 \leq i \leq k\}, \quad (1)$$

$$G_\alpha^k(v) = \{(V_s, E_s) \mid s \in S_k\}, \quad (2)$$

where $e_1 e_2 \dots e_k$ represents the scaled-edge obtained by following an ordered sequence of in-edge ($\leftarrow$) or out-edge ($\rightarrow$) hops from $v$. Specifically:

- $V_s$ consists of all nodes that can be reached from $v$ within $\alpha$ steps of scaled-edge $s$.

- $E_s$ consists of all scaled-edges $s$ from $v$ to these nodes within those $\alpha$ steps.

Consider a GNN model $M$ that learns from a graph $G$ using its adjacency matrix $A$ by aggregating information solely from its out-neighbors. To also learn from the in-neighbors, the model should aggregate information from the transpose of the adjacency matrix, i.e., $A^T$ (Rossi et al., 2024).

An adjacency matrix which encodes scaled-edges is the ordered sequencial multiplication of $A$ and $A^T$. The graph whose structure is represented with the scaled adjacency matrix is a scaled graph.

**Definition 3.6** (Scaled Adjacency Matrix and Scaled Graph)**.** Let $A_k$ denote the set of all $k^{th}$-scale adjacency matrix and $G^k$ denote the set of all $k^{th}$-scale graphs.

$$A_k = \{a_1 a_2 \dots a_k \mid a_i \in \{A, A^T\}, 1 \leq i \leq k\}, \quad (3)$$

$$G^k = \{G^k = (V, \tilde{E}_s) \mid s \in S_k\}, \quad (4)$$

where $\tilde{E}_s$ represents pairwise connections between nodes that are k steps apart in the original graph.

To capture information from $2^{nd}$-scale neighbors, the model should extend its learning to matrices that incorporate both direct and transitive relationships. This involves using matrices such as $AA$, $AA^T$, $A^T A^T$, and $A^T A$ as the scaled adjacency matrix.

**Definition 3.7.** For a node classification task on a graph $G$, we say the task exhibits scale invariance if the classification of a node $v$ remains invariant across different scales of its ego-graphs. Formally, for any $k \geq 1$:

$$f(G_v) = f(G^k(v)), \quad (5)$$

where $f$ is the classification function producing discrete values, $G_v$ is the original ego-graph of node $v$, and $G^k(v)$ is any $k^{th}$-scale ego-graph centered at $v$.

This property implies that the essential structural information for node classification is preserved across different scales of the ego-graph. In other words, the $k^{th}$-scaled ego-graphs should maintain the node classification invariant.

## 4. Proof of Scale Invariance

In this section, we present a proof of scale invariance for Graph Neural Networks (GNNs), exploring the relationship between standard and scaled adjacency matrices in node classification tasks. First, we derive the output of a $k$-layer GCN using the adjacency matrix $A$. We then extend this to scaled adjacency matrices with bidirectional aggregation,

demonstrating that the resulting models are equivalent to dropout versions of lower-scale, bidirectional GCNs that aggregate using both $A$ and $A^T$. The cases of adding self-loops and not adding them are discussed separately. We focus on the Graph Convolutional Network (GCN) model (Kipf & Welling, 2016) as it represents the basic form of neighborhood aggregation.

### 4.1. Preliminaries

Let $X$ denote node features, $A$ denote the adjacency matrix (where an element is 1 if an edge exists and 0 otherwise), $W$ denote a general weight matrix, $D$ denote the degree matrix of $A$, and $I$ denote the identity matrix. For a scaled edge $\hat{e}$ (as defined in Definition 3.4), let $X_{\hat{e}}$ represent the 1-hop neighbors of $X$ through $\hat{e}$, for examaple, $X_I$ denote 1-hop in-neighbors of $X$. $X^k$ denotes representation of nodes after $k$-layer GNN.

**Theorem 4.1.** *The layer-wise propagation of a GCN is:*

* *Without self-loops:* $\sum X_I W$

* *With self-loops:* $\sum X_I W_1 + X W_0$

*Proof.* As outlined in Table A1 (provided in the appendix for completeness), $\tilde{A}$ denotes incidence-normalized A, the layer-wise propagation of a GCN (Kipf & Welling, 2016)is represented as follows:

$$\tilde{A}XW \text{ (no self-loops)}, \quad \widetilde{(A+I)}XW \text{ (with self-loops)}$$

Since incidence normalization corresponds to a component-wise multiplication with the normalization matrix $N$, we have $\tilde{A}XW = (N \odot A)XW$. By the Universal Approximation Theorem (Hornik et al., 1989; Hornik), this is equivalent to $AXW$. Here, $AX$ represents the aggregation of neighbor features, and thus $AX = \sum X_I$, where $I$ represents the 1-hop in-edges. Similarly, $\widetilde{(A+I)}XW$ is $\sum X_I W_1 + X W_0$. □

**Theorem 4.2.** *For all natural numbers $n$, the output of an $n$-layer GCN without self-loops can be expressed as follows:*

$$X^n \approx \sum X_{\underbrace{I...I}_{n}}W,$$

*where $X_{\underbrace{I...I}_{n}}$ denotes neighbours reached by $n$-hop in-edges, $X^n$ denotes representation of nodes after $n$-layer GNN.*

**Theorem 4.3.** *For an $n$-layer GCN with self-loops, the output can be expressed as follows:*

$$X^n \approx \sum X_{\underbrace{I...I}_{n}}W_1 + \sum X_{\underbrace{I...I}_{n-1}}W_2 + ... + X W_{n+1}.$$

The proofs of theorems 4.2 and 4.3 are presented in Appendix D.

Next, we will prove a fundamental property of GNNs for directed graphs: scale invariance. We will demonstrate that when the input graph undergoes scaling transformations, the GCN's output remains unchanged, considering both scenarios—whether or not self-loops are added. This proof highlights that the GNN's architecture inherently preserves its effectiveness and consistency across scaled graph representations, ensuring robust performance in diverse scenarios.

### 4.2. Proof of Scale Invariance in GCN without Self-loops

For different adjacency matrices, the layer-wise propagation rules and $k$-layer outputs are as follows:

#### 4.2.1. SINGLE-DIRECTIONAL AGGREGATION

* $A$ as the adjacency matrix:

  1-layer: $\sum X_I W$; $k$-layer: $\sum X_{\underbrace{I...I}_{k}}W$ for $k \geq 1$

* $A^T$ as the adjacency matrix:

  1-layer: $\sum X_O W$; $k$-layer: $\sum X_{\underbrace{O...O}_{k}}W$

* $AA$ as the adjacency matrix:

  1-layer: $\sum X_{II} W$; $k$-layer: $\sum X_{\underbrace{I...I}_{2k}}W$

* $AA^T$ as the adjacency matrix:

  1-layer: $\sum X_{IO} W$; $k$-layer: $\sum X_{\underbrace{IO...IO}_{k \text{ pairs IO}}}W$

Similar patterns for $A^T A^T$ and $A^T A$.

From above, we can deduce:

1. $k$-layer GCN with $AA$ is equivalent to $2k$-layer GCN with $A$

2. $k$-layer GCN with $A^T A^T$ is equivalent to $2k$-layer GCN with $A^T$

#### 4.2.2. BIDIRECTIONAL AGGREGATION

If the model uses bidirectional aggregation (Rossi et al., 2024), the $k$-layer outputs ($k \geq 1$) are as follows:

* $A$ and $A^T$ as the adjacency matrices:

  $$\sum X_{\underbrace{II...I}_{k}}W_0 + \sum X_{\underbrace{OI...I}_{k}}W_1 + ... + \sum X_{\underbrace{O...O}_{k}}W_k$$

- $AA^T$ and $A^T A$ as the adjacency matrices:

$$\sum X_{\underbrace{IO...IO}_{k \text{ pairs IO}}} W_0 + ... + \sum X_{\underbrace{OI...OI}_{k \text{ pairs OI}}} W_k$$

Similar patterns for $AA$ and $A^T A^T$.

From the above, we can deduce:

1. A $k$-layer GCN with $AA^T$ and $A^T A$ is a dropout version of a $2k$-layer GCN with $A$ and $A^T$. In this context, "dropout" refers to the selective aggregation of information, where specific subsets of neighbors are preserved rather than aggregating information from all neighbors at each step.

2. A $k$-layer GCN with $AA$ and $A^T A^T$ is also a dropout version of a $2k$-layer GCN with $A$ and $A^T$.

Synthesizing Section 4.2.1 and Section 4.2.2, we conclude that all single-directional aggregation models are dropout versions of their bidirectional counterparts. For example, a model using only $A$ corresponds to a bidirectional model with both $A$ and $A^T$, and a model using $AA$ corresponds to a bidirectional model with both $AA$ and $A^T A^T$.

Finally, we can conclude that all models—whether single-directional or bidirectional—are dropout versions of $A$ and $A^T$.

Similar analysis for GCN with self-loops is presented in Appendix C.3.

In conclusion, our theoretical analysis confirms that propagating information through higher-scale adjacency matrices is fundamentally equivalent to applying lower-scale graph operations or their dropout variants. This equivalence not only supports the theoretical validity of scale invariance in graph neural networks but also ensures that the use of multi-scale graphs retains the benefits of invariance across different graph structures.

Furthermore, as undirected graphs can be treated as a special case of directed graphs, where in-neighbors and out-neighbors are identical, the proof of scale invariance extends seamlessly to undirected graph structures. These findings provide a solid foundation for developing more efficient and scalable graph neural network models that leverage multi-scale graph representations.

While we demonstrate the proof of scale invariance specifically for GCN, similar mathematical arguments can be constructed for GraphSAGE and other GNN variants. These findings provide a solid foundation for developing more efficient and scalable graph neural network models that leverage multi-scale graph representations.

Empirical demonstration of scale invariance is presented in Appendix C.5.

## 5. ScaleNet

### 5.1. A Unified Network: ScaleNet

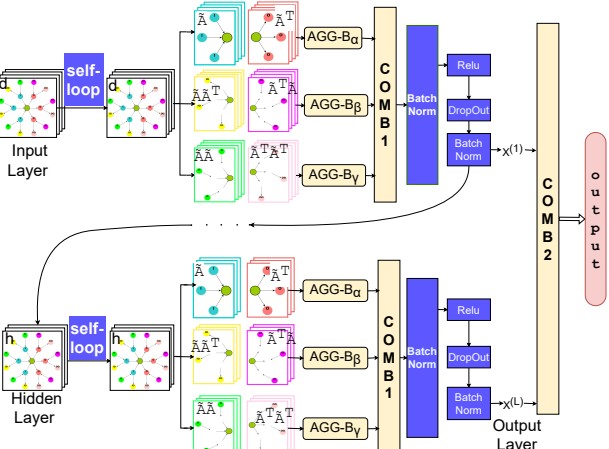

Figure 3: Schematic depiction of multi-layer($L$-layer) ScaleNet with $d$ input channels and $h$ hidden channels. For layer-wise aggregation, the original graph is derived into two $1^{st}$-scaled and four $2^{nd}$-scaled graphs. Three **AGG-B** blocks determine input selection for **COMB1**, which uses either a jumping knowledge architecture (Xu et al., 2018) or addition. **COMB2** represents the fusion of all layers' outputs. (The blue blocks are optional, including self-loop operations, non-linear activation functions, dropout, and layer normalization.)

As discussed in Appendix C.5, heterophilic graphs tend to suffer from performance degradation when aggregating information from scaled graphs in both directions. This limitation causes existing Digraph Inception Networks (Tong et al., 2020a;b) to perform poorly on heterophilic graphs.

To address this issue and accommodate the unique characteristics of different datasets, we propose a flexible combination approach and introduce **ScaleNet**, as illustrated in Figure 3. This approach flexibly synthesizes scaled graphs and optionally integrates components like self-loops, batch normalization, and non-linear activation functions, each of which is tailored to the specific characteristics of the dataset through a grid search of model parameters.

**Bidirectional Aggregation** To exploit scale invariance, we define the bidirectional aggregation function **AGG-B**$_\alpha(M, N, X)$ as follows:

$$(1+\alpha)\alpha \, \mathbf{AGG}(M, X) + (1+\alpha)(1-\alpha) \, \mathbf{AGG}(N, X) \quad (6)$$

The **AGG** function can be any message-passing neural network (MPNN) architecture, such as GCN (Kipf & Welling, 2016), GAT (Veličković et al., 2018), or SAGE (Hamilton et al., 2017). $M$ and $N$ represent pairs of matrices encoding opposite directional edges. The parameter $\alpha$ controls the contribution of matrices $M$ and $N$: $\alpha = 0$ uses only $M$,

$\alpha = 1$ uses only $N$, $\alpha = 0.5$ balances both, and $\alpha = -1$ excludes both.

Given that adding or removing self-loops (Kipf & Welling, 2016; Tong et al., 2020a) can influence the performance of the model, we allow for the inclusion of such options by defining $\tilde{A}$, which can be: (i) the matrix $A$ with self-loops being removed, (ii) the matrix $A$ with self-loops being added, or (iii) the original matrix $A$. The influence of self-loops is shown in Appendix C.4.

This formulation provides a flexible framework for aggregating information from bidirectional matrices, enabling the model to leverage various directional and self-loop configurations to enhance its performance.

Additionally, setting $\alpha = 2$ combines the matrices $M$ and $N$ directly before aggregation, while setting $\alpha = 3$ considers their intersection:

$$\mathbf{AGG\text{-}B}_2(M, N, X) = \mathbf{AGG}(M \cup N, X) \qquad (7)$$

$$\mathbf{AGG\text{-}B}_3(M, N, X) = \mathbf{AGG}(M \cap N, X) \qquad (8)$$

**Layer-wise Aggregation of ScaleNet**  We combine the propagation output from various scaled graphs with the following rule:

$$X^{(l)} = \mathbf{COMB1}(X_1^{(l)}, X_2^{(l)}, X_3^{(l)}, \ldots), \qquad (9)$$

where $X^{(l)}$ represents the updated features after $l$ layers. The function **COMB1** can be realized by the Jumping Knowledge (JK) framework (Xu et al., 2018), or simply by performing an element-wise addition of the inputs.

**Multi-layer ScaleNet**  A multi-layer ScaleNet is then defined as follows:

$$\mathbf{Z} = \mathbf{COMB2}(X^{(1)}, X^{(2)}, \ldots, X^{(L)}) \qquad (10)$$

In this formulation, $L$ layers of the propagation rule are stacked. The function **COMB2** combines the outputs of all layers, which can again be done using the Jumping Knowledge technique; or alternatively, the output from the final layer may be used directly as the model's output.

## 6. Experiments

### 6.1. Datasets

We use six widely-adopted real-world datasets, comprising four homophilic and two heterophilic graph datasets. To ensure consistency and comparability, we maintain the original train/validation/test splits provided by the source datasets. All datasets have 10 splits, except WikiCS, which originally includes 20 splits.

CiteSeer, Cora-ML, and WikiCS are citation networks, while Telegram is a social network. These four datasets

are generally considered to be homophilic (Rossi et al., 2024). Chameleon and Squirrel are webpage networks, and considered heterophilic (Maurya et al., 2022). More details about datasets and experiments are reported in Appendix E

### 6.2. Performance of ScaleNet on Different Graphs

ScaleNet is designed to adapt to the unique characteristics of each dataset, delivering optimal performance on both homophilic and heterophilic graphs. This is achieved through customizable options such as combining directed scaled graphs, incorporating batch normalization, and adding or removing self-loops.

During hyperparameter tuning via grid search, we observed the following key findings:

- **Homophilic Graphs**: Performance improves with the addition of self-loops and the use of scaled graphs derived from opposite directed scaled edges, such as $AA$ and $A^T A^T$.

- **Heterophilic Graphs**: Performance benefits from removing self-loops and utilizing scaled graphs with preferred directional scaled edges, while excluding those based on the opposite directional scaled edges.

- Additional findings:
  - For imbalanced datasets such as the Telegram, incorporating batch normalization significantly improves performance.
  - The CiteSeer dataset performs better with the removal of nonlinear activation functions.

Our unified model, optimized through grid search, reveals the characteristics of different graph datasets and provides a strong basis for model comparison.

Table 4 summarizes the 10-fold cross-validation results. ScaleNet consistently achieves top performance across all six datasets, significantly outperforming existing models on both homophilic and heterophilic graphs.

Model **1**ym assigns 1 to edge weights of model Sym: similarly, model **1**iG and **1**iGi2 are assigning 1 to edge weights of models DiG and DiGib, respectively. Model **1**iGu2 and **1**iGu3 assign weights of 1 to scaled edges, but use union instead of intersection in DiG**i**b, and the last number k denotes the model includes up to $k^{th}$-scale edges, while DiGib only scales up to $2^{nd}$-order. At the end of model name, "ib" would be used interchangeably with "i2". Parameters $\alpha$, $\beta$, and $\gamma$ controlling ScaleNet components: $\alpha$ controls $A$ and $A^T$, $\beta$ controls $AA^T$ and $A^T A$, and $\gamma$ controls $AA$ and $A^T A^T$. Parameter loop is 1 when adding selfloop and 0 when not adding.

Table 4: Node classification Accuracy (%). The best results are in **bold** and the second best are underlined. 10-fold cross validation is used. OOM indicates out of memory on GPU3090 with 24GB of VRAM.

| Type | Method | Telegram | Cora-ML | CiteSeer | WikiCS | Chameleon | Squirrel |
|---|---|---|---|---|---|---|---|
| | MLP | 32.8±5.4 | 67.3±2.3 | 54.5±2.3 | 73.4±0.6 | 40.3±5.8 | 28.7±4.0 |
| | GCN | 86.0±4.5 | 81.2±1.4 | 65.8±2.3 | 78.8±0.4 | 64.8±2.2 | 46.3±1.9 |
| Base models | APPNP | 67.3±3.0 | 81.8±1.3 | 65.9±1.6 | 77.6±0.6 | 38.7±2.4 | 27.0±1.5 |
| | ChebNet | 83.0±3.8 | 80.5±1.6 | 66.5±1.8 | 76.9±0.9 | 58.3±2.4 | 38.5±1.4 |
| | SAGE | 74.0±7.0 | 81.7±1.2 | 66.7±1.7 | **79.3±0.4** | 63.4±3.0 | 44.6±1.3 |
| | MagNet | 87.6±2.9 | 79.7±2.3 | 66.5±2.0 | 74.7±0.6 | 58.2±2.9 | 39.0±1.9 |
| Hermitian | SigMaNet | 86.9±6.2 | 71.7±3.3 | 44.9±3.1 | 71.4±0.7 | 64.1±1.6 | OOM |
| | QuaNet | 85.6±6.0 | 26.3±3.5 | 30.2±3.0 | 55.2±1.9 | 38.8±2.9 | OOM |
| | Sym | 87.2±3.7 | 81.9±1.6 | 65.8±2.3 | OOM | 57.8±3.0 | 38.1±1.4 |
| Symmetric | DiG | 82.0±3.1 | 78.4±0.9 | 63.8±2.0 | 77.1±1.0 | 50.4±2.1 | 39.2±1.8 |
| | DiGib | 64.1±7.0 | 77.5±1.9 | 60.3±1.5 | 78.3±0.7 | 52.2±3.7 | 37.7±1.5 |
| | **1ym** | 84.0±3.9 | 80.8±1.6 | 64.9±2.5 | 75.4±0.4 | 54.9±2.7 | 35.5±1.1 |
| Symmetric | **1iG** | 95.8±3.5 | 82.0±1.3 | 65.5±2.4 | 77.4±0.6 | 70.2±1.6 | 50.7±5.8 |
| (**Ours**) | **1iGi2** | 93.0±5.1 | 81.7±1.3 | 67.9±2.2 | 79.2±0.5 | 58.4±2.5 | 42.7±2.5 |
| | **1iGu2** | 92.6±4.9 | 82.1±1.2 | 67.6±1.8 | 75.6±0.9 | 60.4±2.4 | 40.4±1.8 |
| BiDirection | Dir-GNN | 90.2±4.8 | 79.2±2.1 | 61.6±2.6 | 77.2±0.8 | 79.7±1.3 | 75.6±1.9 |
| **Ours** | ScaleNet | **97.2±2.1** | **82.3±1.1** | **69.1±1.2** | 79.3±0.6 | **80.1±1.5** | **76.0±2.0** |
| | loop_$\alpha, \beta, \gamma$ | 1_0.5,-1,-1 | 1_2,-1,-1 | 1_0.5,2,-1 | 1_0.5,2,-1 | 0_1,1,1 | 0_1,1,1 |

## 6.3. Robustness to Imbalanced Graphs

Table 5: Accuracy (%) on imbalanced datasets (imbalance ratio = 100:1). When accuracy is below 45%, only one split is used.

| Type | Method | Cora-ML | CiteSeer | WikiCS |
|---|---|---|---|---|
| Standard | MagNet | 47.9±5.5 | 29.3 | 62.0±1.5 |
| | Dir-GNN | 41.1 | 25.0 | 62.9±1.4 |
| | DiG | 60.9±1.8 | 36.9 | 72.2±1.4 |
| | DiGib | 55.7±2.9 | 40.4 | 69.8±1.2 |
| Augment | **1iG** | 64.9±4.7 | 42.3 | 71.0±1.5 |
| | **1iGi2** | 61.9±5.7 | 41.5 | 71.0±1.6 |
| | ScaleNet | 60.3±6.7 | 43.1 | 69.4±1.2 |

ScaleNet improves robustness against imbalanced graphs by leveraging multi-scale graphs, similar to data augmentation techniques.

Table 5 indicates that ScaleNet consistently outperforms Dir-GNN and MagNet on imbalanced datasets. The imbalance ratio measures the size disparity between the largest and smallest classes. For homophilic graphs, ScaleNet's advantage stems from its use of higher-scale graphs and self-loops, which enhances its ability to capture essential features that Dir-GNN and MagNet might miss. Conversely, single-scale networks like Dir-GNN (Rossi et al., 2024) and MagNet (Zhang et al., 2021) are prone to incorporate irrelevant nodes due to excessive layer stacking when aggregating information from longer-range nodes.

## 7. Conclusions

We have addressed two critical challenges in Graph Neural Networks: the lack of theoretical support for invariance learning and the absence of a unified model for homophilic and heterophilic graphs. Our work establishes the theoretical foundation of **scale invariance** in graph learning and introduces **ScaleNet**, a unified network architecture that effectively leverages multi-scaled graphs and adaptive self-loops to dynamically handle diverse graph structures.

Through rigorous theoretical analysis, we demonstrate equivalence between Hermitian Laplacian methods and GraphSAGE with incidence normalization and propose efficient alternatives to computationally expensive edge weights in digraph inception networks. Experimental results on six benchmark datasets confirm that ScaleNet achieves state-of-the-art performance across both homophilic and heterophilic graphs while also demonstrating superior robustness to data imbalance.

Our contributions advance the theoretical understanding and practical application of GNNs, offering a unified, efficient, and adaptable framework for graph learning.

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
