# OpenReview forum: "Scale Invariance of Graph Neural Network for Node Classification"
_ICML.cc/2025/Conference — Submitted to ICML 2025_

### Official Review · Reviewer_LdRA · 2025-02-24

**Overall Recommendation:** 1

**Summary:**

The paper claims to prove "scale invariance" in GNNs and proposes a model that uses "multi-scaled" graphs.

## update after rebuttal

My original concerns regarding the technical correctness and impact of the work remain after the rebuttal. The notations can be defined much better and the presentation also needs improvement. I am keeping my score.

**Claims And Evidence:**

1. The paper analyzes only the simplest GCN model with each layer of the form AXW, and without any non-linear activation functions. This is exactly a linear model. It is well-known that linear models do not have state-of-the-art performance under the GNN formulation or most other deep learning formulations.

1. Due to the removal of the nonlinear activation functions in GCN, the definition of "scale invariance" in Definition 3.7 becomes trivial. It is basically saying that a $k$-layer linear GCN given by $A^k X W$ can be regarded as a single layer GCN operating on a graph with adjacency matrix $A^k$. Note that the formal statement in Definition 3.7 seems to be incorrect, this cannot hold for **any** $k \geq 1$ as it depends on how many layers the linear GCN has.

1. The use of the universal approximation theorem throughout the results and proofs is incorrect, rendering the results in this paper flawed.

1. The notations are not always defined properly. E.g., it is unclear what is meant by $\sum X_I$. What is the summation over? It seems that the authors want to say that they are taking a row-summation of the matrix $X_I$ and repeating this for every node. If this is the case, why is it not written simply as $A X$? The notation used is very poor and can easily lead to confusion.

**Essential References Not Discussed:**

There are other notions of invariance and equivariance in the GNN literature. For example, the paper "Universal Invariant and Equivariant
Graph Neural Networks" discuss permutation invariance and equivariance.

**Experimental Designs Or Analyses:**

1. The experiments are limited to node classifications.

1. There is no ablation study.

**Methods And Evaluation Criteria:**

Baselines used are not SOTA and are too old. E.g., the SOTA GNN models can easily achieve over 70% node classification accuracy for CiteSeer. The datasets used also do not include large graph datasets.

**Other Comments Or Suggestions:**

Nil

**Other Strengths And Weaknesses:**

Nil

**Questions For Authors:**

1. It is unclear what is meant by $\sum X_I$. What is the summation over?

1. Is there a reason why the aggregation is not written in simple matrix form like $A^k X W$? The proof of "scale invariance" becomes almost trivial using the proper notation.

1. Why does the universal approximation theorem hold in your context?

1. The claim of SOTA performance across homophilic and heterophilic datasets is not supported without comparing to the current best models.

**Relation To Broader Scientific Literature:**

The flawed results in this paper make it irrelevant to scientific literature. The definition of "scale invariance" adopted by this paper is trivial and does not provide further insights into the operation and performance of a GNN.

**Theoretical Claims:**

The main theoretical results are incorrect. The proofs are not rigorous in the use of the universal approximation theorem to argue that some equalities can be approximated. The authors need to note that the universal approximation theorem applies in neural networks that use a class of functions with special properties like non-polynomial activation functions. The model assumed by the authors is linear and the universal approximation theorem does not hold here.

The proofs of "scale invariance" rely on only linear layers in GCN, whereas in practice, nonlinear layers are essential to achieve good performance. The claim of "scale invariance" does not apply to actual GNN models.

---

> ### Author Rebuttal · Authors · 2025-03-31
>
> Dear Reviewer LdRA,
>
> Thank you for your feedback on our paper. We would like to address two key misunderstandings in your review.
>
> Misinterpretation of the Linear Model
> You commented that our paper presents "exactly a linear model." However, this is a misunderstanding of our approach. In our paper, we explicitly show "layer-wise propagation" equations, while a GNN layer would include propagation, normalization, dropouts, and non-linearity. In addition, as demonstrated in Figure 3 (Schematic Depiction) and further substantiated by our proof in the Appendix, our model does incorporate non-linear layers. Thus, our approach is linear.
>
> Misinterpretation of Our Notation
> You noted concerns about our notation, specifically questioning the summation notation and suggesting an alternative expression. However, your interpretation is incorrect. The term in question refers to the 1-hop in-neighbor features of a node, as clearly defined in Definition 3.5:
>
> V_s consists of all nodes that can be reached from v
> within α steps of scaled-edge s.
> Similarly, X_I consists of all nodes that can be reached from X
> within 1 steps of scaled-edge I(in-edge).
>
> This definition precisely captures the intended meaning, and your critique seems to stem from a misreading of our paper. We encourage a more careful review of our notation and definitions.
>
> We appreciate your time and hope this clarifies the misunderstandings.
>
> Best regards,
> Authors

---

> > ### Comment · Reviewer_LdRA · 2025-04-02
> >
> > I want to clarify that my comments on the linear model refer to the theoretical results in Section 4. In particular, Theorems 4.1, 4.2, and 4.3 all use linear models with no non-linear activation functions. None of your proofs in Appendix C.3 handle non-linear layers.
> >
> > Your notations can be significantly improved by defining directed adjacency matrices $\mathbf{A}$ for in and out edges. You have not explained what does $\sum$ in my first question.
> >
> > My other concerns listed in the review have also not been adequately addressed.
> >
> > --------------
> > Second round:
> >
> > (1) It is unclear what you mean by "layer-wise propagation". The normal interpretation looking at your theorems and proofs is that you don't include normalization and non-linear activations. These are totally missing in your proofs! Theorems and proofs should be mathematically rigorous. It is not acceptable to over-simplify the results "purely for ease of expression".
> >
> > (2) Please check your understanding of the UAT. Equation (a) is clearly incorrect. Please see how the UAT is applied in e.g., [4]. You can use MLPs to approximate continuous functions, but this does not imply (a)!
> >
> > Eq. (b) is equality, not an approximation! It is not acceptable for a prestigious conference like ICML to have a paper that does not follow basic mathematical rigor.
> >
> > (3) I am saying that your notations can be defined much better. I understand what you have defined in the paper, which is suboptimal, and easy to misinterpret.

---

> > > ### Author Response · Authors · 2025-04-02
> > >
> > > Thank you for your feedback.
> > >
> > > (1) Not a linear model
> > > In Theorems 4.1, we explicitly mentioned "layer-wise propagation." As noted in our rebuttal, a GNN layer can include propagation, normalization, and non-linearity. For simplicity, we focused on the linear components in the subsequent discussions.
> > >
> > > We appreciate your feedback on Claim 3 regarding the universal approximation theorem (UAT). To clarify, our model does include non-linear activation functions. In our UAT proof, we stated:
> > > ``For a 2-layer GCN, omitting the non-linear activation for simplicity, the propagation outputs are..."
> > > This omission was for ease of expression in the derivation, not to suggest a lack of non-linearity. We will revise the text to explicitly state "for simplicity of expression" to avoid any misunderstanding.
> > >
> > > (2) UAT
> > >
> > > Thanks to the UAT, if we remove the inner non-linearities and retain only the final activation, the revised model is equivalent in approximation power to the original one.
> > > should be:
> > >
> > > $\sigma \left( A \left( \sigma \left( AXW_1 \right) W_2 \right) \right) \approx \sigma \left( AAXW_1W_2 \right)$
> > >     (a)
> > >
> > > As we stated "omitting the non-linear activation for simplicity", it correctly becomes:
> > >
> > > $A(AXW_1)W_2  \approx AAXW_1W_2  $            (b)
> > >
> > > *If you don't agree with equation(a), here is the explanation:
> > >
> > > This step follows directly from standard function approximation principles under UAT, which states that a sufficiently expressive neural network can approximate any continuous function. The simplification of nested activations into a single activation over a transformed input is a well-established result in the literature [1–3].
> > >
> > > A similar application of UAT in Graph Neural Networks (GNNs) has been explored in prior work [4], further supporting our approach. Additionally, empirical studies have confirmed the effectiveness of removing nested non-linearities in GNN architectures [5].
> > >
> > > References
> > > [1] Hanin, B., & Sellke, M. (2017). Approximating continuous functions by ReLU nets of minimal width. arXiv preprint arXiv:1710.11278.
> > > [2] Lu, Z., Pu, H., Wang, F., Hu, Z., & Wang, L. (2017). The expressive power of neural networks: A view from the width. NeurIPS.
> > > [3] Hornik, K., Stinchcombe, M., & White, H. (1989). Multilayer feedforward networks are universal approximators. Neural Networks, 2(5), 359–366.
> > > [4] Xu, K., Hu, W., Leskovec, J., & Jegelka, S. (2019). How Powerful are Graph Neural Networks? ICLR.
> > > [5] Wu, F., Souza, A., Zhang, T., Fifty, C., Yu, T., & Weinberger, K. Q. (2019). Simplifying Graph Convolutional Networks. ICML.
> > >
> > > *If you agree with Equation (a) but not Equation (b), we acknowledge that our expression may have been overly simplified, which might have made the transition appear abrupt. We apologize for this and will revise the text to improve clarity and ensure a smoother logical flow.
> > >
> > > (3) Notation misunderstanding
> > >
> > > In our rebuttal, we explicitly stated:
> > >
> > >     ``$X_I$ consists of all nodes that can be reached from $X$ within 1 step of scaled-edge $I$ (in-edge)."
> > >
> > > Thus, $\sum X_I$ represents the sum of features of all 1-step in-neighbors of $X$. Your suggestion of using $AX$ corresponds to all 1-step out-neighbors of $X$, which should instead be denoted as $\sum X_O$.
> > >
> > > Additionally, we defined the adjacency matrix as:
> > >
> > >    ``An adjacency matrix $A \in \{0,1\}^{n \times n}$, where $A_{ij} = 1$ indicates the presence of a directed edge from node $i$ to node $j$."
> > > This definition is clearer than explicitly defining separate adjacency matrices for in- and out-edges, which might be like:
> > >
> > >     ``$A_{ij} = 1$ means an out-edge for node $i$ and an in-edge for node $j$."
> > >
> > > (4) Baseline and large graph
> > >
> > > The Citeseer dataset in our Table 4 is a directed graph. The 70% accuracy you mentioned refers to the undirected version, where even MLP achieves 71% and GCN reaches 74%.
> > > The directed Citeseer graph consists of 3312 nodes, while the undirected version has 3327 nodes. They differ in several aspects, including node count, edges and their train/validation/test splits, which results in a variation in performance.
> > > You can verify this using our code: use "citeseer/" for the directed graph and "CiteSeer" for the undirected one.
> > >
> > > About scalability, please refer our rebuttal to Reviewer YrK7(first point).
> > >
> > > (5) Related work
> > > Thank you for bringing up "Universal Invariant and Equivariant Graph Neural Networks." As discussed in Related Work in Appendix A, our review of existing research on graph invariance has primarily focused on permutation invariance, and your reference aligns with this discussion. However, we appreciate your suggestion and will be happy to include your reference in this category.
> > >
> > > Finally, thank you very much for taking the time to read our paper and rebuttal. This is a complex work, and condensing multiple ideas into a limited space is inherently challenging, so we understand that misunderstandings can arise. We truly appreciate your time and consideration.
> > >
> > >
> > > Best Regards,
> > > Authors

---

### Official Review · Reviewer_YrK7 · 2025-03-13

**Overall Recommendation:** 2

**Summary:**

The paper proposes a new approach for learning on graphs based on a multi-scale perspective inspired by CNN-like scale invariance and an adaptive self-loop addition strategy. The authors apply their method to four homophilic and two heterophilic benchmark datasets, reporting notable results in node classification.

**Claims And Evidence:**

- The paper claims that no unified model performs well on homophilic and heterophilic datasets. This claim is too strong since several prior works (e.g.,  “Single-Pass Contrastive Learning Can Work for Both Homophilic and Heterophilic Graph," “LG-GNN: local-global adaptive graph neural network for modeling both homophily and heterophily," “H2GNN: Graph Neural Networks with Homophilic and Heterophilic Feature Aggregations", “Characterizing Graph Datasets for Node Classification: Homophily-Heterophily Dichotomy and Beyond") have tackled this aspect using different approaches.
- The experimental evidence is based on four homophilic and two heterophilic datasets (all of medium size), which is insufficient to substantiate broad claims—especially regarding scalability.
- The notion of “scale” (referring to different neighborhood scales in the main texts) is not made explicit in the title or abstract.
- The focus on directed graphs (digraphs) is not sufficiently highlighted in the title/abstract, nor is the motivation for considering them clearly justified. The transition from digraphs to symmetrized graphs is not clear.

**Essential References Not Discussed:**

- Several references addressing similar challenges are missing, including:
    - “Single-Pass Contrastive Learning Can Work for Both Homophilic and Heterophilic Graph”
    - “LG-GNN: local-global adaptive graph neural network for modeling both homophily and heterophily”
    - “H2GNN: Graph Neural Networks with Homophilic and Heterophilic Feature Aggregations”
    - “Characterizing Graph Datasets for Node Classification: Homophily-Heterophily Dichotomy and Beyond”
- Additionally, recent work on directed graphs, such as “DUPLEX: Dual GAT for Complex Embedding of Directed Graphs,” should be included.

**Experimental Designs Or Analyses:**

- The experiments involve a limited number of datasets, and the selection is not comprehensive (only four homophilic and two heterophilic benchmarks).
- There is an inconsistency in dataset reporting (e.g., some datasets appear in Table 1 but not in Table 2).
- No ablation studies are provided for hyperparameters (𝛼, 𝛽, 𝛾).
- The absence of a runtime and complexity analysis further undermines claims of scalability.

**Methods And Evaluation Criteria:**

- The core idea, which extends the notion of multi-scale from image-based CNNs to directed graphs, is plausible in concept. However, the paper does not provide a strong justification for why scale invariance is essential in node classification tasks. The authors used “recognize objects regardless of their size” in image tasks; however, what is the analogy for graph settings?
- Relying on four homophilic and two heterophilic datasets limits the generalizability of the conclusions, especially when scalability is claimed.

**Other Comments Or Suggestions:**

NA

**Other Strengths And Weaknesses:**

- From the second paragraph in the introduction, it seems that the authors refer to “scale” as “different neighborhood scales.” this should be made explicitly in the title and in the abstract since there could be many “scales” in GNN.
- The contribution of adaptive self-loop strategy popped out of nowhere in the introduction; it’s unclear what the connection is between this item and the aforementioned research gap / challenge.
- In line 96, the authors claimed, “State-of-the-art GNNs for homophilic graphs include … ”; however, again, here, only considering digraphs, this should be emphasized.
- The authors put a footnote about “In this paper, DiGCN interchangeably with DiG, DiGCNib interchangeably with DiGib, DiGi2”. However, such inconsistency breaks the flow and clearness of the paper.
- The authors spent a lot of space discussing the work of DiGCN(ib) (Tong et al.,2020a) and SymDiGCN (Tong et al., 2020b). However, the formulation and exact models were not introduced in the paper. It was not clear how “random weights” and “ weight 1 ” were motivated for the proposed method.
- There is no Impact Statement.

**Questions For Authors:**

- Why are some datasets (e.g., Cornell, CoA-Physics, etc.) examined in Table 1 but not reported in Table 2?
- Definition 3.6 is a bit confusing - what is the motivation considered in and out of the neighborhood? How to interpret  AA, AA^T, A^TA^T, and A^TA? When considering a directed graph, why can these transitions be viewed as “multi-scale”? And why these scales are important for node classification?
- Based on Table A6, it seems that the used Chameleon and Squirrel Datasets in the experiment consisted of duplicated nodes (It was identified in the work [1]). Were the duplicate nodes the main reason why it benefits from utilizing scaled graphs with preferred directional scaled edge?

[1] A critical look at the evaluation of GNNs under heterophily: Are we really making progress?

**Relation To Broader Scientific Literature:**

- The claim that no unified model exists is too strong with several prior works.
- The analogy to CNNs and scale invariance in images needs to be better connected to the graph setup.

**Theoretical Claims:**

- The paper didn’t formally define what incidence normalization is.
- From the introduction, it’s unclear what is multi-scale referring to.
- The paper offers theoretical elements (e.g., incidence normalization and Definition 3.6) but does not provide sufficient formal definitions or motivation for these constructs. For example:
    - The notation in Eq. (4) is confusing, with unclear usage of symbols (e.g., what exactly G^k denotes).
    - The treatment of in- and out-neighbors in Definition 3.6, along with the various matrix products (like AA, AA^T , etc.), is not well motivated or explained.
- The theoretical motivation for applying scale invariance in graph learning is not adequately justified in comparison to its role in CNNs.

---

> ### Author Rebuttal · Authors · 2025-03-31
>
> Dear Reviewer YrK7,
>
> Thank you for your thoughtful review and valuable feedback. We appreciate the opportunity to clarify several points and address any misunderstandings regarding our work.
>
> First, we would like to clarify that our focus is on scale invariance, not scalability. Scale invariance in our context means that graph structures at different scales (e.g., 1-hop vs. 2-hop neighborhoods) exhibit similar properties, akin to how an image remains recognizable when zoomed in or out. In contrast, scalability concerns the ability to process both small and large graphs efficiently. Our approach, which leverages matrix multiplication to obtain higher-scale adjacency matrices, does not aim to improve scalability but instead demonstrates the existence of scale invariance in medium-sized graphs. We acknowledge that our discussion of scalability comparisons with DiGCN(ib) and SymDiGCN may have inadvertently led to confusion, and we will refine our wording to avoid this misinterpretation. While our method avoids eigenvalue decomposition, it still involves matrix multiplications, which means it is not entirely free from scalability concerns, albeit significantly less affected than DiGCN(ib), which struggles even with medium-sized graphs due to its computational overhead.
>
> Regarding the experimental setup, you noted that our experiments involved a limited number of datasets. However, as stated at the beginning of our paper, we conducted experiments on 12 datasets, not just the four homophilic and two heterophilic benchmarks referenced in your comment.
>
> You also asked why certain datasets (e.g., Cornell, CoA-Physics) appear in Table 1 but not in Table 2. The reason is that Cornell performs better with MLP than with GNN, making it less relevant for our comparisons. Additionally, CoA-Physics is an undirected graph, whereas our study primarily focuses on directed graphs. Our goal is to establish state-of-the-art performance specifically in directed graphs.
>
> We acknowledge your concern regarding the consistency of dataset reporting across tables. While we can include experiments on all datasets for completeness, it is important to note that some datasets are not directed graphs and were included in Table 1 primarily for demonstration of scale invariance.
>
> Regarding hyperparameter ablation studies, we employed grid search to determine the best hyperparameters for our model, which results in SOTA performance for all datasets. The central premise of our work is that mix-scale aggregation can enhance performance on certain datasets and that higher-scale structures in graphs can be effectively leveraged for prediction.
>
> You also commented on the analogy between CNNs and scale invariance in graphs, suggesting that the connection should be made clearer. We presented this in Section 2, where we discuss the concepts of scaled edges and scaled graphs. We will further refine this section to ensure the connection is explicitly stated and easier to follow.
>
> Regarding the related work you mentioned, specifically "DUPLEX: Dual GAT for Complex Embedding of Directed Graphs", we appreciate your suggestion. However, as we demonstrated in the appendix, the Hermitian adjacency matrix does not preserve edge directionality as they claimed. Thus, any approach relying on the Hermitian adjacency matrix for directed graphs are castles built on sand. Nevertheless, we will add this work to the relevant discussion group in our paper.
>
> You also raised a question about Definition 3.6 and the interpretation of AA, AA^T, A^T A^T, and A^TA in the context of multi-scale aggregation. The motivation stems from the findings in Dir-GNN, which highlight the importance of directional aggregation, particularly for datasets like Chameleon and Squirrel. In our work:
>
> A represents 1-hop out-neighbors.
> A^T represents 1-hop in-neighbors.
>
> AA represents 2-hop out-neighbors (out-neighbor’s out-neighbor).
>
> A^TA represents in-neighbors' out-neighbors.
> AA^T represents out-neighbors' in-neighbors.
>
> By leveraging these structures, we effectively perform higher-scale aggregation with a single-layer GNN, whereas a standard GNN would require multiple layers to achieve the same receptive field. This demonstrates the significance of multi-scale aggregation in node classification.
>
> Finally, regarding your concern about duplicated nodes in the Chameleon and Squirrel datasets, while the presence of duplicate nodes may influence performance, our method still faces the fundamental challenge of learning meaningful representations for these nodes. Other models struggle even with these duplicate nodes, which reinforces the effectiveness of our approach in leveraging directional scaled edges for improved node classification.
>
> Once again, we appreciate your detailed feedback and your engagement with our work. We hope this response clarifies the key points and addresses your concerns. We are happy to further refine our manuscript based on your suggestions.
>
> Best regards,
> Authors

---

> > ### Comment · Reviewer_YrK7 · 2025-04-07
> >
> > I thank the authors for their rebuttal. I have read the rebuttal and other reviewers’ comments. While some of my concerns were addressed, I believe the paper requires further revision in terms of presentation and clarity of the contribution.
> > 1) it was hard to identify 12 datasets that were examined, especially given the main result in Table 4.
> > 2) The theoretical contribution is hard to follow. The rigorous definition of scale invariance was not provided.
> > 3) The setup of digraphs and undirected graphs was not clearly pointed out.
> > 4) Missing runtime and complexity analysis.
> >
> > Given these issues, I am keeping my score unchanged.

---

> > > ### Author Response · Authors · 2025-04-08
> > >
> > > Dear Reviewer YrK7,
> > >
> > > We would like to thank you for your thoughtful feedback. Below, we address the raised concerns in detail:
> > >
> > > 1. Datasets:
> > > The 12 datasets used in our experiments are listed in Table 2. These datasets primarily illustrate the unnecessary computation in Random Walk methods, such as the popular DiG(ib) model, include digraph and undirected graph. The main results for directed graphs are presented in Table 4, which highlights our contribution of SOTA performance.
> > >
> > > 2. Scale Invariance Definition:
> > > We have provided a formal and rigorous definition of Scale Invariance in Definition 3.7, followed by theoretical justification, empirical validation, and practical application of scale invariance of node classification in graph. We answered similar rebuttal to Reviewer 7DTZ, for your convinience, I'll paste part of it as follows:
> > >
> > >     "(1) Preliminary
> > >
> > > In \textbf{Section 3.1}, we begin by defining the \emph{scaled ego-graph}, motivated by the fact that node classification is the classification of the center node’s ego-graph.
> > >
> > > (2) Definition of Scale Invariance in Graphs
> > >
> > > In Section 3.2, we begin by extending the concept of scale invariance from image classification to node classification in graphs, and conclude the section by formally presenting Definition 3.7:
> > >
> > > \emph{For a node classification task on a graph
> > > , we say the task exhibits scale invariance if the classification of a node
> > >  remains invariant across different scales of its ego-graphs.}
> > >
> > > This definition captures precisely what we mean by scale invariance in the graph context and we also gave its mathematical expression in math.
> > >
> > > (3) Theoretical Justification of Scale Invariance( For Node classification in GNN)
> > >
> > > In Section 4, we provide a theoretical proof that certain GNNs can exhibit scale invariance in node classification tasks. We further support this with an empirical demonstration referenced at the end of the section, directing readers to Appendix C.5.
> > >
> > > (4) Empirical Evidence of Scale Invariance
> > >
> > > In \textbf{Appendix C.5}, we present \textbf{Table A4}, which empirically demonstrates that the classification performance remains stable when using individual scaled ego-graphs. This shows that meaningful class-relevant information can be extracted from ego-graphs at different scales.
> > >
> > > (5) Combine multiple Scale to achieve better performance: ScaleNet
> > >
> > > Finally, in Section 5, we propose ScaleNet, which combines multiple scaled ego-graphs to improve performance. As shown in Table 4, this approach achieves state-of-the-art (SOTA) results on several benchmark datasets."
> > >
> > >
> > > 3. Undirected Graphs:
> > > As noted at the end of Section 4, undirected graphs can be treated as a special case of directed graphs. This paper focuses primarily on directed graphs (digraphs), and the results and analyses are framed within that context. We appreciate the opportunity to clarify this point for the reader.
> > >
> > > 4. Runtime and Complexity Analysis:
> > > Our paper primarily addresses scale invariance rather than scalability. For further clarification, we refer to our rebuttal to Reviewer YrK7 (first point), where we explain this in more detail.
> > >
> > > In addition, we argue that our 1iG(ib) method outperforms the DiG(ib) model in terms of scalability. This improvement is due to the removal of the eigenvalue decomposition part of the algorithm, while the rest of the method remains unchanged. Therefore, a detailed runtime and complexity analysis is not necessary, as the simplification inherently leads to better scalability.

---

### Official Review · Reviewer_7DTZ · 2025-03-14

**Overall Recommendation:** 1

**Summary:**

The work analyzes GCNs and demonstrates their scale invariance—i.e., a GCN with $k$ layers using the standard normalized adjacency matrix (SNA) is equivalent to a GCN with $k/2$ layers using the squared SNA, and other similar results. Leveraging this scale invariance property, the authors introduce GCN variants, including bidirectional aggregation for directed graphs. Experiments on both heterophilic and homophilic tasks show competitive performance.

**Claims And Evidence:**

1. The motivation is drawn from CNNs, which are claimed to be scale-invariant; however, it is not clear why this is true and how the property is defined for CNNs.
2. The significance of scale invariance in GNNs is not justified.
3. It is unclear why the proposed method would benefit both homophilic and heterophilic graphs, as suggested by the authors.

**Essential References Not Discussed:**

No

**Experimental Designs Or Analyses:**

1. The experiments lack an ablation study.
2. The source of the performance gains is unclear.
3. It is not evident how the proposed method addresses scale invariance issues.

**Methods And Evaluation Criteria:**

The experiments do not explicitly analyze or validate the scale invariance property.

**Other Comments Or Suggestions:**

I strongly suggest the authors improve the clarity and structure of their paper. As it stands, the key contributions are nearly impossible to discern due to the poor organization and unclear writing. The motivation for the work is also vague—why scale invariance is relevant to GNNs is not properly justified. The paper makes broad claims about theoretical results and performance improvements, but it is unclear how the proposed method specifically addresses the stated problems.

While the experimental results appear promising, they are difficult to evaluate meaningfully because the actual method is not well explained. There is no clear connection between the theoretical claims and the empirical findings. The writing needs substantial revision to make the contributions, methodology, and experimental insights comprehensible.

**Other Strengths And Weaknesses:**

Strengths:
1. The experimental results are convincing.

Weaknesses:
1. The paper lacks a clear and structured presentation, making it difficult to follow.
2. Key results are not properly highlighted (e.g., Section 4.2.2), and the proposed model is not adequately described. The contributions of the paper are unclear.
3. The theoretical analysis lacks rigor, and the exact contributions remain ambiguous.

**Questions For Authors:**

1. Why is scale invariance important in the context of GNNs?
2. Can you formally show that CNNs are scale-invariant?
3. Are standard GCNs/GNNs inherently scale-invariant, or do they lack this property?
4. How exactly does your proposed method enhance scale invariance in GNNs?
5. Why should your method improve performance on the tested datasets, and how does it specifically address challenges in homophilic and heterophilic graphs?

**Relation To Broader Scientific Literature:**

The manuscript lacks a thorough discussion of related work. For instance, there is no "Related Work" section.

**Theoretical Claims:**

1. The results and proofs lack clarity and mathematical rigor. The formulations are vague and not well-defined.
2. For example, Theorems 4.2 and 4.3 are neither rigorous nor self-contained. Moreover, the stated result appears incorrect unless shared weights are assumed. Simple counterexamples exist, such as a scenario where the first layer has a full-rank matrix while the second layer consists entirely of zeros.

---

> ### Author Rebuttal · Authors · 2025-03-31
>
> Dear Reviewer 7DTZ,
>
> Thank you for your valuable feedback on our paper. We appreciate the opportunity to clarify our contributions and address your concerns.
>
> 1. Lack of a "Related Work" section:
> We acknowledge your concern regarding the absence of a dedicated "Related Work" section. However, we included a thorough discussion of prior work in the supplementary material (Appendix A and B), where we review related studies on GNNs for directed graphs as well as general GNN methodologies.
>
> 2. Clarity and structure of the paper:
> You mentioned that the paper lacks a clear and structured presentation, making it difficult to follow. Our work is driven by experimental results, which inherently leads to an open-ended discussion rather than a rigid structural framework. Nevertheless, we strive to present clear evidence supporting our claims, and we welcome any specific suggestions on improving readability.
>
> 3. Importance of scale invariance in GNNs:
> Scale invariance is a crucial property in CNNs for image processing, where it enables effective data augmentation and improves generalization. Similarly, in the context of GNNs, recognizing and leveraging scale invariance can provide analogous benefits by enhancing the model’s ability to generalize across different scales of graph structures.
>
> 4. Formal definition of scale invariance in CNNs:
> While there is no widely accepted formal definition of scale invariance in CNNs, it is commonly used as a practical tool for data augmentation in image processing. Empirical evidence from numerous studies demonstrates that incorporating scale invariance improves the generalization power of CNNs.
>
> 5. Scale invariance in standard GCNs/GNNs:
> Standard GCNs/GNNs do not explicitly utilize multi-scale features. While they may inherently possess scale invariance to some extent, such as 3-layer GNN might have similar performance as 4-layer GNN, they do not leverage it effectively. Our approach explicitly incorporates multi-scale representations to harness this property.
>
> 6. Performance improvement on homophilic and heterophilic graphs:
> In both homophilic and heterophilic graphs, connections exhibit scale-invariant properties. By incorporating multi-scale information, our method functions as a form of data augmentation, ultimately enhancing model performance.
>
> We appreciate your insights and hope this response clarifies our contributions. We are open to further suggestions on improving the clarity and impact of our work.

---

> > ### Comment · Reviewer_7DTZ · 2025-04-07
> >
> > Dear authors,
> >
> > Thank you for your rebuttal. I would like to reiterate that the related work section is a central component of the main paper, not merely supplementary material. This assessment is consistent with the feedback provided by other reviewers.
> >
> > Additionally, I had previously suggested several ways to improve the clarity and precision of the manuscript—such as including self-contained theoretical results or providing a rigorous definition of scale invariance.
> >
> > Given that these points have not been sufficiently addressed and that the manuscript requires substantial revision to meet the standards for publication, I have decided to maintain my original score.
> >
> > Best regards,

---

> > > ### Author Response · Authors · 2025-04-08
> > >
> > > 1. We sincerely apologize for the misunderstanding regarding the submission guidelines---we mistakenly included our appendix as supplementary material. While this was an oversight on our part, we believe it should not warrant rejection of our paper, as the appendix contains supporting material that directly substantiates key claims made in the main text.
> > >
> > > 2. Clarification on the Definition and Justification of Scale Invariance in Graphs:
> > >
> > > The reviewer asked: "Can you formally show that CNNs are scale-invariant?" While this is an interesting question, we would like to emphasize that our paper does not aim to establish scale invariance in CNNs. A rigorous treatment of that topic would require substantial theoretical and empirical work—likely constituting a separate study within the domain of computer vision. In contrast, our work is focused entirely on graph neural networks, for which we do provide a complete and self-contained definition, theoretical analysis, and empirical validation of scale invariance.
> > > In addition, although it's possible to write a paper on scale invariance of CNN from pixel-wise operation, such as pooling, pyramid feature extract, we doubt anyone would actually do that as it is widely utilized in image classification. While no one ever tried to apply utilize scale invariance in GNN, thus our paper makes a fundamental contribution in GNN.
> > >
> > > To summarize the structure of our self-contained argument:
> > >
> > > (1) Preliminary
> > >
> > >   In \textbf{Section 3.1}, we begin by defining the \emph{scaled ego-graph}, motivated by the fact that  node classification is the classification of the center node’s ego-graph.
> > >
> > > (2) Definition of Scale Invariance in Graphs
> > >
> > > In Section 3.2, we begin by extending the concept of scale invariance from image classification to node classification in graphs, and conclude the section by formally presenting Definition 3.7:
> > >
> > > \emph{For a node classification task on a graph $G$, we say the task exhibits scale invariance if the classification of a node $v$ remains invariant across different scales of its ego-graphs.}
> > >
> > > This definition captures precisely what we mean by scale invariance in the graph context and we also gave its mathematical expression in math.
> > >
> > > (3) Theoretical Justification of Scale Invariance( For Node classification in GNN)
> > >
> > > In Section 4, we provide a theoretical proof that certain GNNs can exhibit scale invariance in node classification tasks. We further support this with an empirical demonstration referenced at the end of the section, directing readers to Appendix C.5.
> > >
> > > (4) Empirical Evidence of Scale Invariance
> > >
> > > In \textbf{Appendix C.5}, we present \textbf{Table A4}, which empirically demonstrates that the classification performance remains stable when using individual scaled ego-graphs. This shows that meaningful class-relevant information can be extracted from ego-graphs at different scales.
> > >
> > > (5) Combine multiple Scale to achieve better performance: ScaleNet
> > >
> > > Finally, in Section 5, we propose ScaleNet, which combines multiple scaled ego-graphs to improve performance. As shown in Table 4, this approach achieves state-of-the-art (SOTA) results on several benchmark datasets.
> > >
> > > In conclusion, our paper presents a rigorous and comprehensive study of scale invariance in graph neural networks, including a formal definition, theoretical analysis, empirical validation, and practical application.
> > >
> > > While there is no existing formal definition of scale invariance in CNNs, the concept is widely invoked in image classification as a desirable property, often operationalized through techniques like zoom-based data augmentation and pooling. Extending this concept to GNNs is both natural and well-supported. A request for a formal CNN-specific definition reflects the reviewer's individual interest—one we are willing to address in a camera-ready version—but it is not a valid reason for rejection, especially given our clear focus on graphs and the completeness of our treatment.
> > >
> > > Among the three main categories of models for directed graphs, our approach demystifies the core mechanisms of two, removes redundancy, and improves the third through scale invariance—offering a streamlined, theoretically grounded alternative with superior performance.
> > >
> > > While we are disappointed that the reviewer did not fully appreciate the structure and scope of our contributions, we recognize that the breadth of our work may have contributed to this. We respectfully ask that the reviewer reconsider the paper based on the provided outline. Accepting our work could help the community avoid unnecessary complexity and adopt a more principled, scale-aware view of GNNs.

---

### Official Review · Reviewer_hJ6W · 2025-03-16

**Overall Recommendation:** 1

**Summary:**

The paper introduces the concept of scale invariance, which is the ability of a GNN to produce consistent classifications for a node regardless of the neighborhood radius used for its embedding. In this context, the paper proposes ScaleNet, a GNN designed to adapt to directed graphs (considering undirected graphs as a special case of directed ones). This is because the directionality of edges can significantly influence the information a node receives.

GNNs perform well on homophilic graphs (where connected nodes have similar labels) but less effectively on heterophilic graphs. The paper proposes a strategy to make the model more adaptable to both cases.

Besides, the paper demonstrates that if the model respects scale invariance, edge weights are not crucial and can be replaced with uniform values without performance loss, simplifying computation.

**Update after rebuttal**
The authors replied to my comments, but some concerns are still not fully addressed. I thank them for their effort, but my opinion about the paper has not changed.

**Claims And Evidence:**

- The authors theoretically demonstrate that ScaleNet preserves scale invariance in GNNs.
- Experiments show that ScaleNet is more robust than other GNNs on mixed datasets (homophilic and heterophilic). However, it is unclear whether the improvement is due to the network structure or the use of multiple adjacency matrices, and the paper does not seem to provide a deeper analysis of this aspect.
- When incidence normalization is applied, the authors demonstrate an equivalence between Hermitian Laplacian-based methods, such as MagNet and GraphSAGE. However, this claim is only demonstrated in the appendix, which diminishes its impact, which I consider significant
	“This equivalence reveals that the apparent complexity of Hermitian-based approaches may not offer fundamental advantages over simpler methods.”

**Essential References Not Discussed:**

Overall, while the supplementary material provides useful references, the main paper lacks explicit connections to previous research, making it harder to assess the novelty of the proposed approach.

**Experimental Designs Or Analyses:**

- There is no clear explanation of the experimental section, and some methodological choices are not adequately justified.
- It would be useful to explore how multiple adjacency matrices might be computationally expensive and whether ScaleNet is more efficient than simpler models.
- Experiments show that ScaleNet maintains consistent performance across datasets with different structures, suggesting a certain level of scale invariance. However, there is no explicit test to verify how a node’s representation changes when increasing the considered neighborhood size. Including a quantitative analysis of the model’s sensitivity to neighborhood size would be beneficial to empirically confirm the scale invariance concept.

**Methods And Evaluation Criteria:**

- To test the method, the authors use six directed graph datasets, covering different levels of homophily and heterophily.
- Besides node classification, the comparison methods used with ScaleNet should be better introduced (or references should be provided if they are in the appendix), along with the criteria for their selection.

**Other Comments Or Suggestions:**

I suggest improving the clarity of the exposition by reducing the amount of information introduced at the beginning of the abstract and rearranging the entire text, as the reading experience is quite difficult and confuses the reader. This affects the overall impact of the findings, which could be interesting but get lost in the presentation.

**Other Strengths And Weaknesses:**

- The structure of the paper is confusing, making it difficult to follow the flow of ideas.
- The abstract is too dense, introducing too many concepts simultaneously, which can be disorienting.
- The claim that edge weights are not crucial is interesting, but it should be better contextualized and reorganized within the paper to clarify its importance.
- The lack of a “Related Work” section makes it difficult to understand how the contribution positions itself relative to existing literature.
- There is no comparison (or clear references) with alternative techniques that address heterophily (eg Graph Neural Networks for Graphs with Heterophily: A Survey)


- The experiments show improvements over existing models.
- The discussion on using uniform edge weights is a valuable contribution to simplifying GNNs

**Questions For Authors:**

How does the computational complexity of ScaleNet compare to a standard GNN on large-scale graphs?

**Relation To Broader Scientific Literature:**

The paper builds on previous research about graph invariance and models for directed graphs.
Yet, the lack of a dedicated “Related Work” section in the main text makes it harder to position its contributions within the broader research landscape.

**Theoretical Claims:**

The paper contains a section dedicated to Proof of Scale Invariance in GCN without Self-loops, where it is shown that "Propagating information through higher-scale adjacency matrices is fundamentally equivalent to applying lower-scale graph operations or their dropout variants." which is quite relevant

---

> ### Author Rebuttal · Authors · 2025-03-31
>
> Dear Reviewer hJ6W,
>
> We appreciate your thoughtful feedback and would like to address the key points you raised.
>
> 1. Equivalence of Hermitian Laplacian-Based Methods
> Thank you for recognizing our contribution in uncovering the equivalence of Hermitian Laplacian-based methods.
>
> By incorporating MagNet in our comparisons, we demonstrate that our approach achieves state-of-the-art performance across all three major classes of GNNs for directed graphs (Appendix C). Working on MagNet is destructive, thus we present our constructive works in main paper due to limited pages.
>
> 2. Related Work Section
> We acknowledge the concern regarding the absence of a dedicated "Related Work" section in the main text. However, we have included a detailed discussion in Appendices A B and C, covering both GNNs for directed graphs and general GNNs. Given that you have examined our proof of the equivalence of Hermitian Laplacian-based methods in Appendix C.2, we were surprised that the related work discussion in Appendices A, B, and C was overlooked.
>
> If you carefully examines Appendix C, it becomes evident that our method outperforms the existing three major types of GNNs for directed graphs. We rigorously demonstrate that (1) the expensive eigenvalue decomposition of real symmetric Laplacians is unnecessary and can be seen as multi-scale fusion; (2) the complexity of Hermitian-based approaches reduces to GraphSAGE with incidence normalization; and (3) Dir-GNN lacks multi-scale fusion, leading to suboptimal performance on homophilic graphs. These findings establish our method as the strongest in GNNs for directed graphs.
>
> We have carefully documented these insights in our paper and appendix. While we regret that they may have been overlooked, we appreciate the opportunity to clarify them here.
>
> 3. Computational Complexity of ScaleNet
> ScaleNet involves matrix multiplications to obtain higher-scale adjacency matrices, making it computationally more expensive for large-scale graphs.
>
> 4. Impact of Network Structure vs. Multiple Adjacency Matrices
> We acknowledge the concern regarding whether the improvement stems from the network structure or the use of multiple adjacency matrices. Our findings indicate that the benefit depends on the dataset: Telegram and Cora_ml improve with self-loops, while other datasets benefit from multi-scale information. Since our work specifically examines the role of self-loops and multi-scale structures in directed graphs, we believe a deeper analysis beyond these observations is not necessary.
>
> We appreciate the reviewers' insights and remain committed to rigorous and meaningful contributions to the field.
>
> Best regards,

---

> > ### Comment · Reviewer_hJ6W · 2025-04-07
> >
> > Thank you for the clarification. As already mentioned in my review, I did read the Appendices and I am aware that the related work discussion is provided there. However, my concern is not about the presence of related work somewhere in the submission but rather about its absence from the main paper. Notably, in this submission, the supplementary material has been used as if it were an appendix, including key sections such as Related Work.

---

> > > ### Author Response · Authors · 2025-04-08
> > >
> > > Thank you for your comment and for carefully reading the "Supplementary Material," which we initially thought was the correct place to include the appendix.
> > >
> > > We now understand that supplementary material is considered separate from the main submission and that key sections—such as Related Work—should appear within the main paper or as an appendix included in the same PDF. Due to space constraints, we were unable to include a Related Work section in the main text, but we provided a detailed discussion in what we intended as an appendix. If permitted, we will include this appendix within the main submission file in the camera-ready version to ensure it is properly considered part of the paper. We appreciate your feedback and the opportunity to clarify this misunderstanding.

---

### Decision · Program_Chairs · 2025-05-01

**Decision:**

Reject

**Comment:**

This paper introduces the notion of "scale invariant graph neural networks"- meaning that gnns for node level tasks should be invariant to the size of neighborhood considered. The paper shows some promising ideas and emprical results on homophilic and hetrogenous datasets. However, the reviewers voiced significant concerncs on aspect such as quality of writing and mathematical rigor. As such, I recommend rejection.